# Direct observation of exciton–exciton interactions

Jakub Dostál[1], Franziska Fennel [2,3], Federico Koch[1], Stefanie Herbst[2], Frank Würthner [2,3]
Tobias Brixner [1,3]

Natural light harvesting as well as optoelectronic and photovoltaic devices depend on efficient transport of energy following photoexcitation. Using common spectroscopic methods, however, it is challenging to discriminate one-exciton dynamics from multi-exciton interactions that arise when more than one excitation is present in the system. Here we introduce a coherent two-dimensional spectroscopic method that provides a signal only in case that the presence of one exciton influences the behavior of another one. Exemplarily, we monitor exciton diffusion by annihilation in a perylene bisimide-based J-aggregate. We determine quantitatively the exciton diffusion constant from exciton–exciton-interaction 2D spectra and reconstruct the annihilation-free dynamics for large pump powers. The latter enables for ultrafast spectroscopy at much higher intensities than conventionally possible and thus improves signal-to-noise ratios for multichromophore systems; the former recovers spatio–temporal dynamics for a broad range of phenomena in which exciton interactions are present.

[1] Institut für Physikalische und Theoretische Chemie, Universität Würzburg, Am Hubland, 97074 Würzburg, Germany. [2] Institut für Organische Chemie, Universität Würzburg, Am Hubland, 97074 Würzburg, Germany. [3] Center for Nanosystems Chemistry (CNC), Universität Würzburg, Theodor-Boveri-Weg, 97074 Würzburg, Germany. Correspondence and requests for materials should be addressed to
F.Wür. (email: wuerthner@uni-wuerzburg.de) or to T.B. (email: brixner@phys-chemie.uni-wuerzburg.de)

Most conventional time-resolved spectroscopy experiments are carried out with the premise that the optical excitations of molecular electronic transitions (excitons) introduced by an excitation pulse are independent of each other. Then the observed ensemble dynamics arise as the sum over individual constituents. This requires sufficiently low laser intensities such that the generated excitons are well separated spatially. In highly-interconnected systems with closely packed absorbers such as given in natural or artificial light-harvesting systems it is challenging to meet this condition. The exciton delocalization over several molecular constituents[1,2] together with the exciton mobility across the system highly increases the chance that the presence of one exciton influences the behavior of another one. In this case we speak about exciton–exciton interaction (EEI). The unobscured single-exciton signal is detectible only at very low excitation intensities with an inferior signal-to-noise ratio, while high-intensity signals contain artefacts from EEI effects such as exciton–exciton annihilation.

In some multichromophore systems the interactions between excitons might be directly connected to their functional properties and are therefore of direct scientific interest. For example, on a sunny day, the majority of solar energy harvested by a photosynthetic organism is dissipated in the form of heat due to the congestion of the light-harvesting apparatus[3–5]. In this situation, multiple excitons interact because the presence of one exciton influences the dynamics and energy-transfer pathways of the other one: The energy of one exciton is being slowly processed in the reaction center, while the other excitons have to be quickly dissipated out of the apparatus in order to prevent photodamage. The direct detection of EEI thus might unravel the details of the photoprotective mechanisms.

In systems with high exciton mobility the EEI can provide insight into single-exciton transport properties, since an interaction of two initially well separated non-interacting excitons can arise after time, i.e., after spatial propagation. Even though it might be necessary to set the excitation intensity higher than the typical operation conditions of the studied systems in order to prepare enough excitons capable of interacting, the EEI-specific signal still carries information about microscopic processes in the studied system that would be relevant for low-light conditions.

In conventional third-order coherent two-dimensional (2D) spectroscopy one often studies the nonlinear signal emitted along directions $-\mathbf{k}_1 + \mathbf{k}_2 + \mathbf{k}_3$ and $\mathbf{k}_1 - \mathbf{k}_2 + \mathbf{k}_3$, where $\mathbf{k}_1$, $\mathbf{k}_2$, and $\mathbf{k}_3$ are the wave vectors of the three incident pulses[6–15]. The resulting "absorptive" 2D spectra can be interpreted as a generalization of transient absorption with additional resolution of the excitation energy and primarily reflect the single-exciton dynamics. Some information on double excitations is contained in the third-order double-quantum coherence signal $\mathbf{k}_1 + \mathbf{k}_2 - \mathbf{k}_3$[16–20].

For a full dynamical measurement of EEI, higher-order methods are required since none of the third-order methods can initially prepare two excitons in the sample and monitor their interaction with time. One possibility for that would be to simply increase the excitation power such that additional terms $\mathbf{k}_1$–$\mathbf{k}_1$ and $\mathbf{k}_2$–$\mathbf{k}_2$ arise along the $-\mathbf{k}_1 + \mathbf{k}_2 + \mathbf{k}_3$ and $\mathbf{k}_1 - \mathbf{k}_2 + \mathbf{k}_3$ directions. The newly arising signals are present for high pump powers in standard transient absorption as well as in absorptive 2D spectroscopy and overlay the absorptive features, which makes the isolation of the EEI signal complicated if possible at all. Thus avoiding these multiexciton contributions typically requires extremely low light levels in multichromophore systems[21,22].

Another way of revealing multiexciton signals is the "multiple population-period transient spectroscopy" (MUPPETS)[23,24] detecting fifth-order signals in $\pm(\mathbf{k}_{1'} - \mathbf{k}_{1''}) \pm (\mathbf{k}_{2'} - \mathbf{k}_{2''}) + \mathbf{k}_3$ direction. Here all five pulses are focused into the sample from a different direction and pulses numbered by the same numeral

are simultaneous. This technique isolates the fifth-order signal but does not further distinguish between two-exciton and pump–dump processes[25]. It also delivers a signal for two-level systems that cannot contain two excitons. Similar limitations are inherent to other current fifth-order techniques[26,27].

Our approach consists of analyzing and experimentally implementing the $-2\mathbf{k}_1 + 2\mathbf{k}_2 + \mathbf{k}_3$ and $2\mathbf{k}_1 - 2\mathbf{k}_2 + \mathbf{k}_3$ directions, which, as we show, are specific to EEI. This signal contribution has been calculated numerically by Brüggemann and Pullerits[28] for the photosynthetic Fenna–Matthews–Olson (FMO) complex demonstrating that the structure of two-exciton states could be revealed. Here, we show in detail that the nature of the signal is more general. It appears only if one exciton influences the behavior of another one and it vanishes in the case of non-interacting excitons. We utilize the signal properties to introduce "exciton–exciton-interaction two-dimensional (EEI2D) spectroscopy" that is designed to work at high laser intensities and is sensitive to the interaction between two excitons. We demonstrate how the EEI2D spectra of a perylene bisimide J-aggregate can be experimentally acquired and how they can be used to characterize exciton diffusion properties. Lastly we show that the pure single-exciton dynamics can be reconstructed from simultaneously detected EEI2D and absorptive 2D spectra.

## Results

**Theoretical concept.** Before providing a formal treatment of EEI2D, we describe the essence of the technique on a more intuitive level. Coherent 2D spectroscopy is a generalization of transient absorption spectroscopy in which the sample is excited by a coherent pair of pump pulses instead of a single one[8,13]. The spectrum of such a coherent pulse pair necessarily contains interference fringes, the density of which can be reliably controlled by the time separation between the constituting pulses (so-called coherence time). All processes subsequently observed in transient absorption can thus be related to the excitation of the sample at a set of specific frequencies (the maxima of the fringes). By systematic scanning of the delay between the pulses, we periodically modulate the excitation of the sample at any given frequency and the excitation at each frequency is modulated with a different rate. If the transient change of absorption scales linearly with excitation power, we can uniquely relate each component of the transient absorption signal to a single excitation frequency ("single-excitation-frequency component") by observing its periodical appearance and disappearance in the overall signal. In practice this is done by taking the Fourier transform with respect to the coherence time. The resulting signal is named third-order "absorptive" 2D spectrum and appears in the spectral region covered by the pump pulses. Another signal component appears around the origin of the excitation frequency axis (corresponding to that part of the signal not sensitive to scanning the coherence time).

Multiphoton processes in the sample (such as, e.g., any interaction between two excited systems or pump-dump process) make the scaling of the transient change of absorption non-linear with excitation power, which has to be described by (predominantly) fifth-order perturbation theory. In this case, if the 2D spectrum is acquired as described above, Fourier transforming over the coherence time cannot separate the single-excitation-frequency components of the transient absorption signal from each other anymore. The changes in the 2D spectrum are twofold: Firstly, the absorptive 2D spectrum is overlaid with an additional fifth-order signal, which complicates its physical interpretation. Secondly, an additional signal appears in the new region of twice the excitation frequency, where no single-exciton third-order contribution is present. It can be

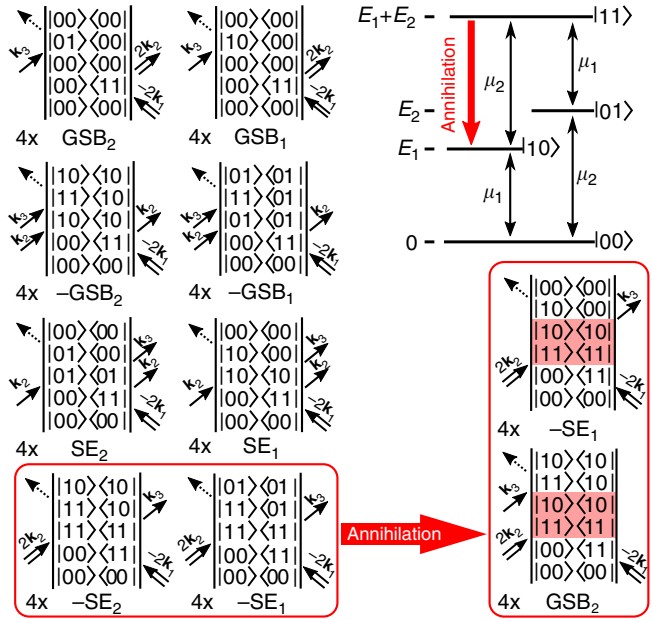

**Fig. 1** Double-sided Feynman diagrams for EEI2D spectroscopy. Top right: Energy scheme of a non-interacting pair of two-level systems (black) with an idealized annihilation channel in the case of interaction (red). Symbols $E_1$ and $E_2$ label the excitation energies of individual subsystems and symbols $\mu_1$ and $\mu_2$ their respective transition dipole moments. Left two columns: Feynman diagrams of non-interacting molecules attributed to ground-state bleach (GSB) or stimulated emission (SE) signals of the individual subsystems (denoted by the numerical index). The diagrams perfectly compensate each other. In the case of interaction, the annihilation process replaces the two lowest diagrams (red frame) with a new pair (bottom right) leading to a non-zero EEI signal. The indicated diagram signs are given by the parity of number of interactions acting on the right side (odd: negative, even: positive) as we neglect the additional overall −1 factor coming from the fifth order of the perturbation theory. Each diagram is present four times (as indicated) since each of the double interactions allows for two different time orderings

characterized by $-2\mathbf{k}_1 + 2\mathbf{k}_2 + \mathbf{k}_3$ and $2\mathbf{k}_1 - 2\mathbf{k}_2 + \mathbf{k}_3$ phase-matching relations and is of high interest since it carries solely information about two-exciton processes. This signal provides valuable direct insight into EEIs fully separated from single-exciton dynamics.

After this qualitative explanation, we now analyze more formally the general properties of the fifth-order $-2\mathbf{k}_1 + 2\mathbf{k}_2 + \mathbf{k}_3$ ("rephasing") and $2\mathbf{k}_1 - 2\mathbf{k}_2 + \mathbf{k}_3$ ("non-rephasing") EEI signals. Assuming the impulsive limit and neglecting non-resonant interactions, we discuss a perturbative treatment of the density-matrix time evolution via double-sided Feynman diagrams[6]. Each diagram as given in Fig. 1 graphically represents a single possible time-ordered sequence of interactions of the system (described by the density matrix) with the electric field at a given order of the perturbation theory. The current state of the system is represented by the matrix element located in the center of the diagram, the electric field interacting with the system's "ket" or "bra" vector is given by arrows located on the left or right side, respectively. Time flows from the bottom to the top.

An isolated two-level system is free of any EEI signal as no Feynman diagrams with a double excitation step exist. Let us now consider systems with more than two energy levels in which $-2\mathbf{k}_1 + 2\mathbf{k}_2 + \mathbf{k}_3$ and $2\mathbf{k}_1 - 2\mathbf{k}_2 + \mathbf{k}_3$ signals can occur[28]. A non-interacting pair of molecules can be described as a pair of two-level systems (Fig. 1, top right). The energy-level structure

contains a single common ground state |00>, two singly-excited states representing excitation of either the first (|10>) or the second (|01>) molecule, and one doubly-excited state |11> representing the simultaneous excitation of both molecules. The particular feature of EEI2D spectroscopy is that processes which are described with Feynman diagrams containing the state |11> can be resolved as a distinctive feature at twice the excitation energy, whereas conventional third-order techniques are limited to diagrams with single-chromophore excitations only (or, to be more precise, where the full ensemble dynamics can be described as if only a single chromophore had been excited).

As a two-level system does not yield any EEI signal, a pair of independent two-level systems also cannot give any EEI signal. This intuitive result is confirmed by analyzing the Feynman diagrams of the total system [Fig. 1, left two columns for rephasing signals, and Supplementary Figures 1 and 2 for non-rephasing and oscillatory signals]. For each positive contribution there is another contribution of equal magnitude but opposite sign leading to overall perfect cancellation as expected.

The assumption of independent population dynamics of both subsystems (decay of states |01> and |10>) necessarily fully determines the lifetime and decay pathways of the state |11>. For example, in the simplest case of single-exponential population dynamics the decay of state |11> follows the combined rate of both constituting subsystems. This physical requirement assures the perfect cancellation of the overall EEI signal at any time after excitation in case of non-interacting subsystems.

In case the excitons interact with each other the population dynamics of |11> is modified and no longer given by the behavior of the individual constituting subsystems. We rather observe a new, independent physical process, and the associated set of Feynman diagrams does not cancel anymore. The detailed properties of the resulting EEI signal depend on the nature of the multiexciton process. As a useful example we further investigate idealized exciton–exciton annihilation. However, the derivation can be repeated in analogous form for other types of interaction between excitons. In our annihilation model we assume that excitation of both molecules leads to the complete disappearance of one of the excitons. In the energy-level scheme this is reflected by the population relaxation from |11> to one of the singly-excited states, say |10>. This process converts the negative stimulated emission Feynman diagrams of the annihilated exciton into positive ground-state bleach diagrams of the same exciton (replacement shown by red boxes in Fig. 1). Adding up all diagrams reveals that the total EEI signal surviving the mutual cancellations now consists of the stimulated emission and the ground-state bleach (i.e., the full species-associated spectrum) of the annihilated exciton. Note that the excited-state absorption of the disappearing exciton, if present in its experimental species-associated spectrum, would be part of the appearing EEI signal as well, although it is not part of the here-presented model. The total signal subsequently decays with the lifetime of the remaining exciton.

The initial excitations can be far apart from each other and EEI might be mediated via some third system located in between. For example, in a one-dimensional J-aggregate excitons can migrate along the chain. When they meet, one of them is annihilated[29,30] and the EEI signal appears with a time scale corresponding to exciton diffusion (1–100 ps). In reality, annihilation occurs via a highly excited state that temporarily carries the energy of both excitons. This might manifest in an additional signature in the EEI spectra. However, in our case we assume a lifetime of below 150 fs as observed for a similar aggregate[31], such that a population does not accumulate and the signal will be weak, if detectable at all.

Another possible source of an EEI2D signal is a direct double excitation of the same quantum system[28], which is allowed by its

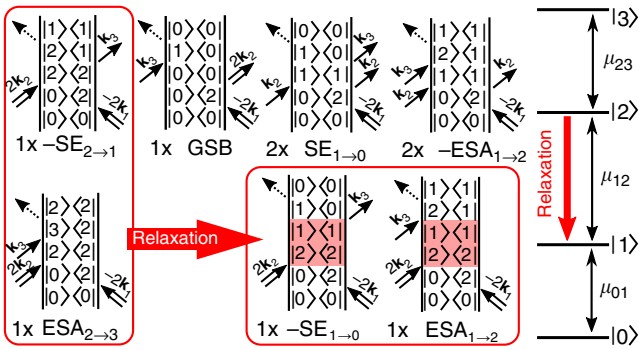

**Fig. 2** Origin of the direct-double-excitation EEI signal. Right: Energy scheme of a four-level system with indicated relaxation from its second excited state (red arrow). Left: Double-sided Feynman diagrams of the rephasing EEI signal for direct double excitation. Some of the diagrams are present twice (as indicated) due to multiple possible time orderings of two interactions with the same laser pulse. The diagrams are classified as stimulated emission (SE), ground-state bleach (GSB), and excited-state absorption (ESA) with numerical indices that label the states of the final transition. The population relaxation replaces two of the diagrams by a new pair as indicated by the red boxes

highly excited state lying approximately twice higher in energy than the single excited state as illustrated on a four-level system in Fig. 2. Since the excitons interact immediately after (double) excitation, the EEI2D signal is present from the very beginning. Its appearance is given by the interplay of several contributions of both negative and positive signs (Fig. 2). The positions and intensities of the contributions in the EEI2D spectra are given by the energy level spacing and strengths of their transition dipole moments.

In molecular aggregates we consider the one-, two-, and three-exciton states as relevant. For one-dimensional aggregates consisting of $\delta$ coupled monomers it can be approximately shown (see Supplementary Note 2) that the magnitude of the transition dipole moments from the one- to the two-exciton state, $|\mu_{12}|$, and from the two- to the three-exciton state, $|\mu_{23}|$, scales with respect to the transition dipole moment from the zero- to the one-exciton state, $|\mu_{01}|$, in the following way: $|\mu_{12}|^2/|\mu_{01}|^2 = 2(\delta-1)/\delta$ and $|\mu_{23}|^2/|\mu_{01}|^2 = 3(\delta-2)/\delta$. Using these relations, the integrated intensity of the EEI2D spectrum perfectly cancels to zero while the EEI2D spectrum itself is expected to be non-zero, since the mutually canceling contributions do not perfectly overlap in frequency.

The fast population relaxation from higher-exciton states into the single-exciton states, which typically happens in aggregates on a 100-fs time scale[32], leads to a new set of diagrams (as indicated by the red arrows and red boxes in Fig. 2). When this process is finished, the EEI signal consists of ground-state bleach, stimulated emission, and (negative) excited-state absorption contributions that remain visible in the EEI2D spectra until the remaining exciton disappears. The net integrated signal of the relaxed aggregates is positive. We thus expect to see a ~100-fs rise of the integrated EEI2D signal followed by its decay on the nanosecond time scale in case multi-exciton processes contribute.

The total EEI2D signal in a molecular aggregate is given by a combination of both EEI processes, i.e., considering the diagrams in Figs. 1 and 2 as well as Supplementary Figures 1 and 2. Relative intensities of both effects are given by the microscopic properties of the sample and the experimental conditions.

**Experiment**. We implement a partially collinear pump–probe geometry and use phase cycling via femtosecond pulse shaping of

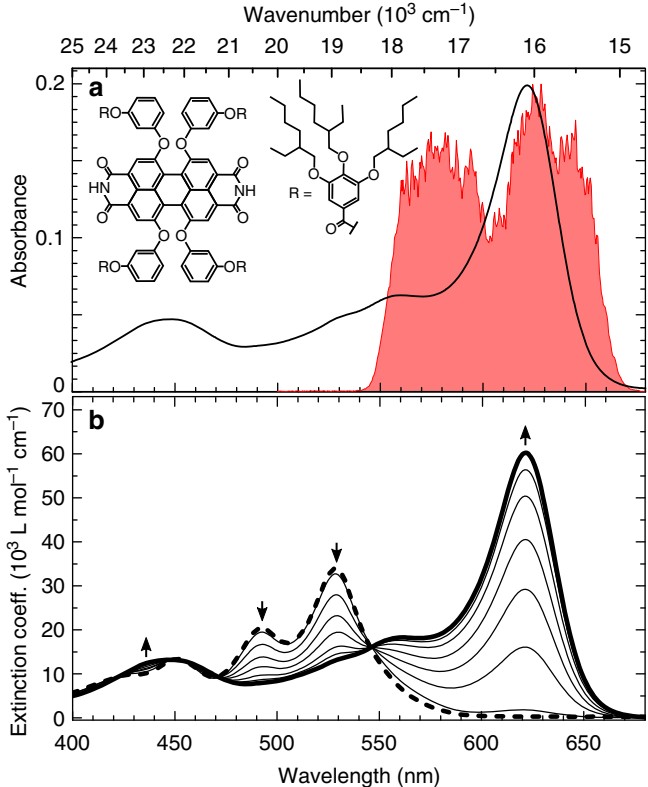

**Fig. 3** Absorption properties of MEH-PBI aggregates. **a** Molecular structure of monomeric MEH-PBI compound (inset), actual absorbance spectrum of MEH-PBI aggregates in MCH (with a molecular concentration of $1.5 \times 10^{-4}$ M) as used for EEI2D experiments and laser spectrum of pump pulse (red). **b** Concentration-dependent absorption spectra of MEH-PBI in MCH at 20 °C in the concentration range from $6 \times 10^{-6}$ to $1.7 \times 10^{-4}$ M. Arrows indicate spectral changes with increasing concentration

the pump pulses to extract the relevant EEI2D contribution, rather than detecting along the phase-matching directions $-2\mathbf{k}_1 + 2\mathbf{k}_2 + \mathbf{k}_3$ and $2\mathbf{k}_1 - 2\mathbf{k}_2 + \mathbf{k}_3$. This has the advantage that we acquire the conventional absorptive 2D and the EEI2D spectra simultaneously, located around the single and the double excitation laser frequency, respectively.

We carry out EEI2D spectroscopy on a sample of a quasi-one-dimensional J-aggregate consisting of a core-substituted perylene bisimide dye (MEH-PBI)[33] in methylcyclohexane (MCH). In nonpolar solvents MEH-PBI dyes self-assemble by a concerted interaction of H-bonds and π–π-stacking interactions into extended one-dimensional nanofibers. The monomer structure of MEH-PBI is shown in Fig. 3a and its linear absorption spectrum in Fig. 3b (black thick dashed line). Upon increasing concentration (Fig. 3b, black thin lines) self-assembly of MEH-PBIs is observed, leading to J-aggregates (black thick line). Several well-defined isosbestic points suggest the presence of a two-state equilibrium system (monomer and J-aggregate) which is almost completely shifted to the aggregate state at concentrations above $1 \times 10^{-4}$ mol L$^{-1}$. The pronounced bathochromic shift from 529 (monomer) to 621 nm (J-aggregate) corroborates the presence of a J-coupled exciton system[34]. The pump laser spectrum (red) covers the absorption maximum around 16,000 cm$^{-1}$ (625 nm) and is shown together with the absorbance of the actual sample in Fig. 3a.

Experimental signals (Fig. 4) appear in two distinct regions of the excitation axis. The absorptive 2D spectrum is visible around the linear absorption maximum of 16,000 cm$^{-1}$ and exhibits the shape typical for molecular aggregates[35–39]. The dominant

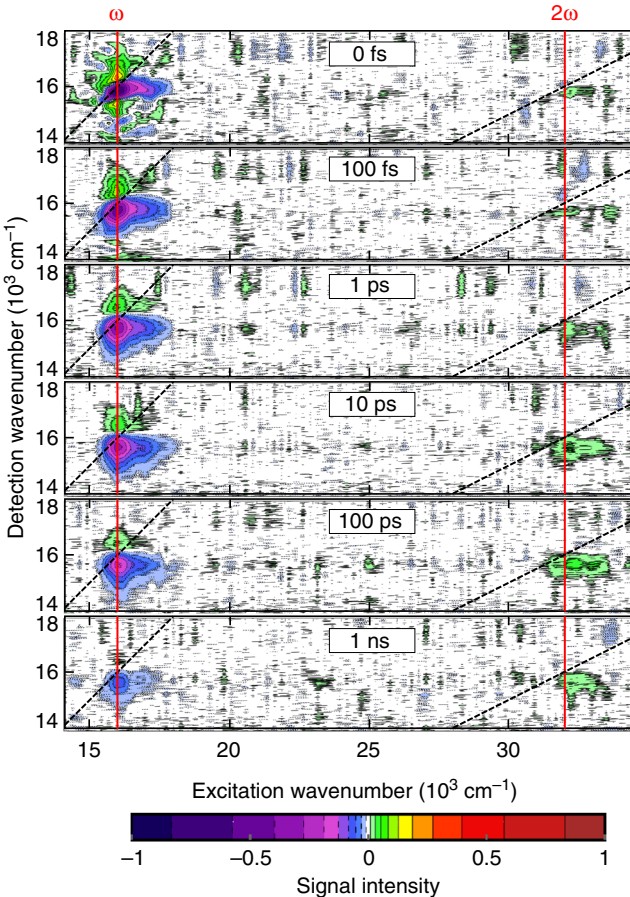

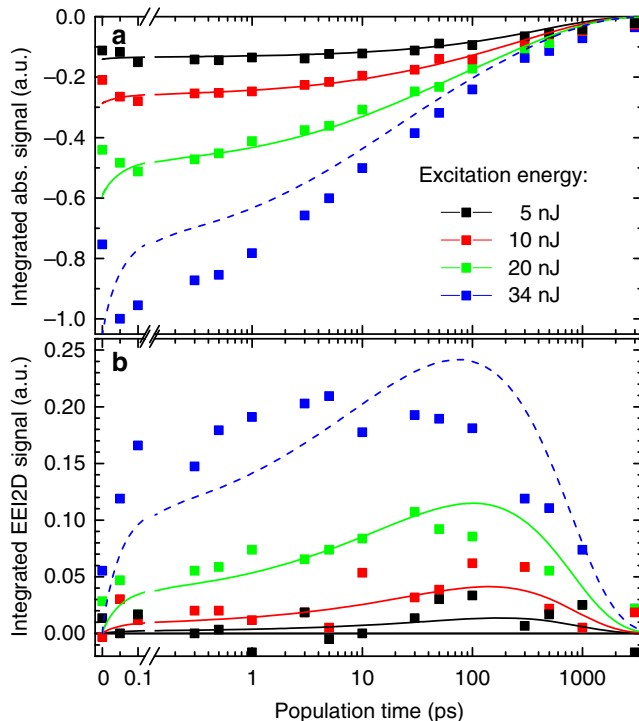

**Fig. 5** Integrated time evolution of the 2D spectra. The two graphs show the time evolution of **a** the integrated experimental absorptive 2D and **b** the EEI2D spectra for various excitation intensities (symbols). The 5, 10, and 20 nJ experimental data sets are compared with their simultaneous best fit of the theoretical model (solid lines). The 34 nJ dataset is compared with the same model prediction (dashed curve) using identical fitting parameters

**Fig. 4** Two-dimensional spectra of MEH-PBI aggregates. The conventional absorptive 2D spectrum occurs near the maximum of the laser spectrum at 16,000 cm$^{-1}$ (red vertical line labeled as ω) and the EEI2D spectrum near 32,000 cm$^{-1}$ (red vertical line labeled as 2ω). The spectra are shown for several population times as indicated. Excitation pulse energy was set to 10 nJ. Contours are drawn at fractions of 0.01, 0.03, 0.05, 0.08, 0.13, 0.19, 0.27, 0.4, 0.57, and 0.83 of the maximal signal amplitude. To stress the connection with transient absorption, we employ the same sign convention and plot a decrease of absorption as negative and an increase as positive

negative peak located slightly below the spectral diagonal (black dashed) combines ground-state bleach and stimulated emission. At higher detection frequencies (above the diagonal), weaker positive excited-state absorption is discernible.

The EEI2D spectrum around 32,000 cm$^{-1}$ lies outside the spectral bandwidth of the pump pulse. Since each order of the perturbative expansion contains an additional complex factor of $i$, the individual contributions in the fifth-order EEI2D signal appear with an opposite sign than in the third-order absorptive 2D spectrum. The EEI2D signal is dominated by stimulated emission and ground-state bleach located at the same detection frequency as in the absorptive 2D spectrum. Excited-state absorption is, due to its relatively small intensity, not recognizable in Fig. 4. However, it can be glimpsed in the spectra taken with higher excitation intensities (Supplementary Note 3 and Supplementary Figure 4). The spectral shape of the EEI2D spectrum is thus fully in accordance with the theoretical prediction stating that it represents the species-associated transient spectrum of the annihilated exciton. The total energy of the interacting exciton pair determines the signal position along the horizontal excitation axis. Since diffusion brings random pairs of excitons together, the EEI2D spectrum is elongated along the excitation axis.

In agreement with theoretical expectations, the absorptive 2D signal is strongest directly after excitation while the EEI2D signal appears at later times reaching its maximum at about 100 ps. A qualitatively similar behavior is apparent for higher excitation intensities (Supplementary Figure 4), though with a different time scale for reaching the signal maximum. Also, at higher powers, some part of the EEI2D signal appears directly after excitation due to the direct double excitation of the same aggregate domain as this contribution gains relevance with increasing excitation power.

For further analysis we plot the time evolution of the integrated absorptive 2D (Fig. 5a) and EEI2D signals (Fig. 5b) for varying excitation pulse energies. As already apparent from the 2D spectra, the absorptive signal decays monotonically to zero with increasing population time as the number of excitons gradually decreases due to the single-exciton dynamics (i.e., relaxation back to the ground state) and their mutual annihilation. With increasing pump intensity, the signal decays faster as annihilation gains relevance.

The integrated EEI2D signal slowly rises for low excitation intensities (Fig. 5b, black and red points) reaching its maximum around 100 ps. This signal is predominantly caused by the diffusive motion that randomly brings pairs of excitons together. For higher excitation intensities (Fig. 5b, green and blue points), direct excitation of the two-exciton state leads to an additional contribution within the temporal resolution of the experiment (~100 fs).

In order to confirm the attribution of the EEI2D signal to exciton–exciton annihilation, we conducted the same experiments on a monomeric dye (Nile blue in ethanol) under identical conditions (Supplementary Note 3 and Supplementary Figure 5).

In this case we observe no diffusion-mediated EEI2D signal because the mean molecular separation (at 0.2 mM concentration) is too large for EEI, but a small contribution from direct double excitation exists.

## Discussion

In conventional transient absorption or absorptive 2D spectroscopy, third-order and fifth-order contributions overlap. While excitation-density-dependent transient absorption, analogous to data displayed in Fig. 5a, has been analyzed previously to reveal the exciton diffusion constant[29,30,40,41], the disentanglement of single-exciton and annihilation contributions might be ambiguous for data evaluation.

We model the experimental data taking into account three dynamical processes: natural decay with a time constant of $\tau$, diffusion-driven exciton–exciton annihilation, and direct excitation of the two-exciton state. In accordance with previous models the individual steps of exciton transfer are considered to be incoherent such that the overall motion can be described as 1D diffusion characterized by diffusion constant $D$[29,31,40,42–46]. We assume that as soon as two excitons meet, one of them is annihilated. Coherence properties of excitons are taken into account to the extent that the minimal transfer distance is assumed to be the delocalization length $\delta$.

In order to fully describe the evolution of the EEI2D signal we had to include effects of direct population of the two-exciton state. This effect has not been considered in the previous analysis of transient absorption data[29,31,40,42–46] since it is not apparent without direct separation of the EEI part of the signal. The probability for directly populating a two-exciton state increases with the exciton delocalization length and determines the relative intensity of the corresponding EEI2D signal. The fast relaxation from the two-exciton state (either populated via direct excitation or arising from exciton diffusion and subsequent EEI) into the one-exciton state is below our experimental resolution and is approximated by assuming a fixed 50 fs time constant. We obtain analytic solutions (see Supplementary Note 2 for derivation) for the time evolution of the integrated EEI2D and the absorptive spectra, respectively,

$$S_{\text{EEI}} = \frac{1}{3}\alpha\left(1 - \frac{1}{2}\delta\frac{n_0}{N}\xi\right)\xi\,\frac{\exp\left(-\frac{t}{\tau}\right)\text{erf}\left(\sqrt{\frac{t}{\tau}}\right)}{\frac{\beta}{\left(1 - \frac{1}{2}\delta\frac{n_0}{N}\xi\right)\xi} + \text{erf}\left(\sqrt{\frac{t}{\tau}}\right)} \tag{1}$$
$$+ \frac{1}{3}\alpha(\delta - 1)\frac{n_0}{N}\xi^2\left(\exp\left(-\frac{t}{\tau}\right) - \exp\left(-\frac{t}{50\,\text{fs}}\right)\right),$$

$$S_{\text{abs}} = -\alpha\xi\left(1 + \xi\left(\frac{1}{2}\delta - 1\right)\frac{n_0}{N}\right)\exp\left(-\frac{t}{\tau}\right) + 3S_{\text{EEI}}. \tag{2}$$

Equations (1) and (2) appear to be quite complex. Note, however, that despite that appearance, they contain only four independent fitting parameters, by which all time traces for the different excitation powers are fitted simultaneously: $\alpha$, $\beta$, $\delta$, and $\tau$ (see Supplementary Note 2 for definitions). Briefly, parameter $\alpha$ is the proportionality factor between the number of excitons in the sample and the observed signal and does not characterize the exciton dynamics directly. The average number of excitons that every exciton can meet and annihilate with, $\beta^{-1}$, is used to determine the diffusion constant $D$. The excitation powers are linearly related to the lowest 5 nJ excitation intensity by the factor $\xi$ known from experiment. The number of absorbed photons per constituting monomer at 5 nJ excitation, $n_0/N$, is given by the experimental conditions as $0.0053 \pm 0.0006$.

The integrated time evolution of the absorptive 2D and EEI2D spectra for excitation energies of 5, 10, and 20 nJ can be well described using one set of fitting parameters (black, red, and green solid lines, respectively, in Fig. 5). The match between

**Table 1 Physical parameters characterizing the exciton in MEH-PBI aggregate determined from fitting the experimental spectra**

| | |
|---|---|
| Exciton diffusion constant $D$ | $(3.1 \pm 1.1)$ nm$^2$/ps |
| | $(27 \pm 8)$ sites$^2$/ps |
| Number of excitons per their diffusion length for the 5 nJ excitation $\beta^{-1}$ | $(1.1 \pm 0.1)$ |
| Exciton delocalization length $\delta$ | $(3.1 \pm 0.6)$ nm |
| | $(9 \pm 2)$ sites |
| Exciton diffusion length $(2D\tau)^{1/2}$ | $(70 \pm 10)$ nm |
| | $(200 \pm 30)$ sites |
| Exciton lifetime $\tau$ | $(730 \pm 90)$ ps |

relative signal intensities is excellent and the shape of data and model coincide concluding that the model is well suited to describe the underlying physical mechanisms. The extracted relevant physical parameters characterizing the exciton dynamics in the aggregate are summarized in Table 1. Using the lattice constant of 0.34 nm[33] we determine an exciton diffusion of $(3.1 \pm 1.1)$ nm$^2$/ps, resulting in an exciton diffusion length $L_D = (2D\tau)^{1/2} = (70 \pm 10)$ nm that is well in the range reported for similar aggregates[29]. The exciton delocalization length over $(9 \pm 2)$ sites is larger than anticipated.

The data at highest intensity (34 nJ) go beyond our model. Using the same best fit parameters (Fig. 5, blue dashed line) the experimental data is not described in satisfying manner. We attribute this discrepancy to three- and higher-exciton states. The observed deviations show the expected tendency as increasing the direct multiple exciton population leads to a faster appearance of the integrated EEI2D signal (Fig. 5b). Due to the fast decay of the multiple-exciton states, less excitons remain that contribute to annihilation, and therefore the experimental signal is weaker at later times compared with the model prediction.

We thus demonstrated that meaningful values of fitting parameters can explain the observed strength and time evolution of the EEI2D signal, which validates the EEI2D spectroscopy concept. The detailed description of complicated condensed-phase quantum-mechanical processes in molecular aggregates is beyond the scope of the rather simplistic model. Most notably the dipole approximation used to estimate the oscillator strengths of higher energy states is not fully adequate for tightly packed systems[47]. This, together with neglecting the influence of higher-excited monomer states[48], results most likely in the retrieved rather high exciton delocalization length. Furthermore, a detailed description of early-time dynamics would require addition of purely quantum-mechanical processes such as, e.g., dynamical exciton localization[49,50] or ballistic propagation[51]. We hope that the present work will stimulate detailed theoretical treatment of such phenomena in future studies.

The presence of annihilation in the absorptive 2D spectra becomes most apparent by normalizing the integrated time evolution by the excitation intensity (Fig. 6a). The decay of the signal is in all cases non-exponential and accelerates with the increasing initial exciton population. Using EEI2D spectroscopy, however, we can reconstruct the interaction-free (without annihilation, direct double excitation, and further interaction contributions that were not considered in the model) dynamics by subtracting three times the integrated EEI2D signal from the absorptive transient signal (see Supplementary Note 1 and Supplementary Figure 3 for a derivation). Despite the noise level of the reconstructed curves being rather high, their amplitude and shape (Fig. 6b) resemble well the single exponential decay with a lifetime of $(730 \pm 90)$ ps as determined from the fitting model. While

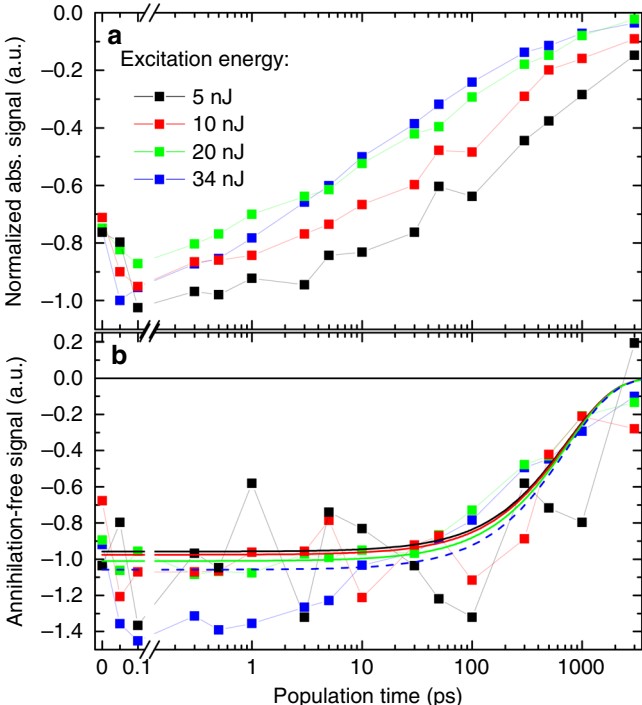

**Fig. 6** Reconstruction of interaction-free signal. **a** Time evolution of integrated experimental absorptive 2D spectra normalized by the excitation intensity. **b** The reconstructed interaction-free decay of the individual signals. The experimental data (squares connected by lines for a better visual guidance) are compared to the model-predicted interaction-free decay (solid and dashed lines in **b**)

the curves in Fig. 6a deviate systematically from each other, the remaining deviations in Fig. 6b constitute purely random noise (except for the short-time behavior of the blue curve), validating experimentally the correctness of our analysis. Note that the reconstruction of an annihilation-free third-order signal works independently of any particular assumed kinetic model. The deviations of the high-energy blue curve are most likely caused by experimental imperfections.

Our results indicate that it is indeed possible to reconstruct the interaction-free decay (which is of third order in perturbation theory) despite the non-negligible presence of fifth-order contributions. The corresponding experimental conditions can be realized with higher excitation power compared with the commonly required case of the dominating third-order single-exciton term, which opens a way to investigate the single-exciton effects of highly interconnected systems.

In summary, we proposed and analyzed fifth-order $-2\mathbf{k}_1 + 2\mathbf{k}_2 + \mathbf{k}_3$ and $2\mathbf{k}_1 - 2\mathbf{k}_2 + \mathbf{k}_3$ nonlinear signals as a basis for measuring interactions between excitons. An EEI2D signal arises when the presence of one exciton influences the time evolution of another one. We derived detailed properties of the EEI2D signal in the case of exciton–exciton annihilation as well as for a direct population of the two-exciton state. We demonstrated EEI2D on a molecular J-aggregate revealing the exciton diffusion constant and the exciton delocalization length. We also reconstructed the annihilation-free single-exciton dynamics from the experimental data without the necessity for measuring at low excitation intensities, which simplifies dynamic studies of multi-chromophore systems. EEI2D spectroscopy should be useful beyond studying annihilation dynamics as it can be applied in all situations with interacting excitations present in a quantum system. Examples include non-geminate recombination in solar-cell

materials, excessive energy dissipation in photosynthetic systems, or multiphoton chemical reactions.

## Methods

**Sample.** MEH-PBI was synthesized as described previously[33]. MEH-PBI aggregation was achieved by a cooperative nucleation-elongation self-assembly process in the non-polar solvent methylcyclohexane. The resulting sample exhibited an optical density of 0.2 at 622 nm in a 0.2-mm flow cell corresponding to a concentration of $1.47 \times 10^{-4}$ M of monomeric constituents. The sample was pumped through the cuvette for the entire time of the experiment with low flow velocity in order to avoid molecular alignment. As a control sample an ethanol solution of Nile blue of 0.3 optical density at 628 nm was used at the same experimental conditions. All experiments were conducted at room temperature.

**Experiment.** An argon-filled (1 bar) fused-silica hollow-core fiber (UltraFast Innovations) was pumped by an 800-nm Ti:Sapphire amplifier (Spitfire Pro, Spectra Physics) at 1-kHz repetition rate. The resulting white-light continuum was partly compressed by a set of chirped mirrors. One part of this beam was directly focused (intensity FWHM: ~40 μm) into the sample and served as a probe. The remaining fraction served as the pump beam and was guided through a grism compressor and an acousto-optic programmable dispersive filter (Dazzler, Fastlite) that reduced its spectrum as indicated in Fig. 3 and compressed it to 11 fs. The Dazzler was also used to generate the pulse pairs and to implement phase cycling. The pump beam was focused (intensity FWHM: ~170 μm) and overlaid in the sample with the probe at an angle of 2°. The relative polarization of the two beams was set to magic angle and the delay (population time) was adjusted by a mechanical delay line (M-IMS600LM, Newport). In order to measure optical density changes of the sample without unwanted contributions from sample scattering and fluorescence, both the pump and probe beams were periodically blocked by a pair of optomechanical choppers (MC2000, Thorlabs). Four configurations were recorded (pump and probe, only pump, only probe, and dark-camera signal) and cycled on a 5-ms (five-shot) basis. In addition, the data were repeatedly acquired with four different combinations of electric-field phases of the individual pump pulses (each dataset is in the following denoted by the pair of phases enclosed in brackets) and combined according to the following scheme: $(0, 0) - (0, \pi/2) - (\pi/2, 0) + (\pi/2, \pi/2)$. This "phase cycling" led to further suppression of the unwanted contribution. In addition, it doubled the relative intensity of the EEI2D part of the acquired 2D spectrum with respect to its absorptive 2D part (see Supplementary Table 1).

The experiments were conducted with excitation powers of 5, 10, 20, and 34 nJ (determined in a situation where the two excitation pulses temporally overlap and fully constructively interfere), which corresponds to fractions of $(0.0053 \pm 0.0006)$, $(0.011 \pm 0.001)$, $(0.021 \pm 0.002)$, and $(0.036 \pm 0.004)$ absorbed photons per monomeric constituent (see Supplementary Methods for details).

In order to generate 2D spectra the delay between the two pump pulses (coherence time) was scanned from 0 to 64.6 fs with steps of 0.38 fs providing a resolution of 260 cm$^{-1}$ along the excitation axis. The resolution was 20 cm$^{-1}$ along the detection axis as determined by the spectrometer characteristics (Acton SpectraPro 2558i assembled with camera Pixis 2 K, Princeton Instruments). The spectra were not corrected with respect to the remaining chirp present in the probe pulse that was about 100 fs across the spectral region of the detected transient absorption signal. This distorts the initial time evolution of the acquired data.

**Data availability.** The data that support the findings of this study are available from the corresponding author upon reasonable request.

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

## Acknowledgements

We received funding by the DFG within the Research Unit "Light-Induced Dynamics in Molecular Aggregates (FOR 1809)" (T.B. and F.W.), the ERC within Consolidator Grant "MULTISCOPE" (T.B.), and the State of Bavaria within the "Solar Technologies Go Hybrid (SolTech)" research program (T.B. and F.W.).

## Author contributions

J.D. proposed, designed and performed the EEI2D experiment. F.F. and F.K. performed supporting experiments. J.D. and F.F. developed the theoretical model and analyzed the data. S.H. synthesized the sample and optimized the self-assembly conditions for J-aggregate formation. T.B. and F.W. supervised the work and discussed the results. J.D. and T.B. wrote the paper with input from all coauthors.
