## [Peer Review File · Nature Communications]

Reviewer #1 (Remarks to the Author):

Comments for authors

In this manuscript, the authors present a novel interpretation of 2D pump-probe spectroscopy, where the use of high powers gives access to fifth order phenomena. The primary results claimed are (i) the characterization of exciton-exciton interactions and (ii) the possibility of reconstructing the third order response 'cleaned up' by fifth-order contamination.

Overall, the paper is well written, the contents are presented in an ordered way, and almost all the results are credible. The paper undoubtedly deserves publication, but I am not sure that in the present form the paper is suitable for Nature Commun.

I have two main concerns.

1) First, the paper is very technical, and I believe that in few parts the reader could be easily lost. Below more details about that:

1.a) the paper is not self-consistent and heavy use of SuppInfo is necessary to follow the contents described in the main text. I understand the space restrictions of Nature Comm, but maybe a paper of this kind should be published somewhere else in a full article form, with part of the SuppInfo integrated into the main text.

1.b) the 'Theoretical concept' section starting on page 3 is very technical. Indeed, while the Feynman diagrams formalism is nowadays well-known and largely employed in the ultrafast spectroscopy community, I believe that this part results pretty heavy and awkward to follow for a non-specialist reader and for the broad readership of Nature Comm. This is a second reason why I would suggest the publication on a more specialized journal, which would also give the possibility of integrating part of the SuppInfo in the main text.

1.c) Along the same line, the discussion on page 5, where the authors discuss annihilation, exciton-exciton interactions and how this is manifested or not in 2D signals and the possible cancelation of diagrams is very cumbersome.

2) About the reliability of the conclusions and claims:

2a) While the authors thoroughly convinced me that EEI2D could really capture qualitatively exciton-exciton annihilation and characterize exciton diffusion and dynamics, I am not sure that these data

can be used quantitatively to extract the exciton parameters reported in Table 1. The calculation of these parameters is based on a model (very complicated, I have to say) relying on a numerous set of fitting parameters. I am not convinced that, also considering the quality of the data and the noise level, there is just a unique set of parameters fitting well the data. The author should comment more on this point and provide additional proofs.

2b) I am even less convinced about the correction of the total signal to retrieve the 'pure' third order signal subtracting the fifth order contributions (claim 2).

The error bar in Figure 5b is so big that it is almost impossible to assess the correctness of the model. In other words, I think that probably a completely different model will perfectly match the data as well. The authors comment about 'small deviations,' but these deviations are in the order of the 80% of the signal!

Also, I do not agree with the conclusion on page 13 where the authors write: 'The match between relative signal intensities is excellent and the shape of data and model coincide concluding that the model is well suited to describe the underlying physical mechanisms. The predicted and reconstructed interaction-free dynamics (Figure 5b) closely resemble each other as well.'

The idea of merely subtracting 3x the fifth order contribution from the overall signal appears to be very naive and not really complying with the complex dynamics characterizing a complex system at high energy. Without considering that at high powers the perturbative approach on which Feynman diagrams formalism is based is not entirely valid anymore.

Reviewer #2 (Remarks to the Author):

In this work, the authors describe a new experiment called “exciton-exciton-interaction two-dimensional spectroscopy” which is a fifth order spectroscopy technique that can detect exciton-exciton interactions. In a sense this is rather idiosyncratic, because multiple exciton systems are unlikely to exist naturally. For example, in biological antenna systems natural sunlight is of low intensity and incoherent, and hence is unlikely to produce these types of phenomena.

Nevertheless, this work is still fundamentally interesting and is an important advance on third order spectroscopy, because it enables one to study situations in which two excitations exist in a system simultaneously. This may therefore open up new directions of research.

Consequently I believe it is publishable in Nature Communications. However, as I explain below, I do have some issues with its present form.

The paper is well written with significant supplementary information. The authors explain the process of interest well, within terms of the traditional double-sided time ordered diagrams. The authors apply the methodology to J-aggregates. This seems an ideal application for this studying exciton-exciton interactions, because of the high conjugation and aggregation properties of these molecules.

In figure 3 of the manuscript, the authors identify the EEI2D signal in the spectra, showing that exciton-exciton interactions can indeed be seen. The authors are also able to extract some dynamical information about the exciton-exciton annihilation process, which is indeed a very exciting result! This shows important promise for future applications for this technique.

The weakness of the paper is the modeling. In my opinion, I think it could be removed and the paper would still be an important contribution. The problem is that exciton-exciton interactions described in this paper are complicated condensed phase quantum mechanical processes. As the authors admit, the theoretical fits do not satisfactorily fit the experimental data. The authors attribute this to “three- and higher-exciton states”. However, I have significant issues with using simplistic phenomenological kinetic models to describe complex quantum processes. There are numerous approximations made that the authors are either not aware of, or do not wish to acknowledge. First, they make the dipole approximation, which may well be unsuitable for exciton-exciton interactions, especially in the condensed phase. (See *J. Chem. Phys.* 107, 5374 (1997) for details.) The “Initial Exciton Distribution” section of the SI could be more rigorously linked to coherent photonic states (see *Molecular Quantum Electrodynamics* by Craig and Thirunamachandran or *The Quantum Theory of Light* by Rodney Loudon). The authors should also look at more rigorous quantum dynamical studies performed on J-aggregates that include explicit quantum processes (*J. Chem. Phys.* 137, 034109 (2012) and *New J. Phys.* 16 (2014) 113067), which describe dissipative and QED effects, respectively.

In summary, I believe that this is an important experimental advance, and is publishable in *Nature Communications*. However, I don't see what the modeling section adds. It certainly doesn't help to identify a mechanism, but seems more like a curve fitting exercise, which doesn't really work. Ideally the authors should use a more rigorous model that accounts for quantum mechanical effects in J-aggregates. I do understand that such a model is outside the scope of this paper, so the modeling section could either 1) be removed, or 2) deficiencies should be fully acknowledged in the main paper (not in the SI). I believe that this is very important, because in this field especially, it is easy to fall into the trap of over-interpreting experimental data based on questionable theoretical models. I strongly believe it is important that future studies address this head-on for the good of the field.

Reviewer #3 (Remarks to the Author):

Dostál et al. detail the innovation of a new ultrafast multidimensional optical spectroscopic technique, exciton-exciton-interaction two-dimensional spectroscopy (EEI2D), to isolate inter-exciton interactions and elucidate exciton diffusion and delocalisation length scales of a model J-aggregate dye system. These key parameters are important to gaining a more fundamental understanding of exciton transport which is key to the primary steps of natural photosynthesis and the field of photovoltaic materials, especially those involving organic photoabsorbing layers.

This well written and thorough manuscript details both the theoretical derivation behind EEI2D and its experimental implementation. These 5th order experiments have been handled with due diligence and care, power dependencies have been performed, alongside control experiments on annihilation free samples. Together this provides a strong rationale and proof that the experimental observations of the J-aggregate MEH-PBI are indeed correct.

The paper will have an impact on how the ultrafast spectroscopic community investigate exciton-exciton interactions, and I expect after the paper's publication and dissemination, the EEI2D technique will be adopted by other leading laboratories around the globe.

It is my very firm recommendation that this manuscript should be accepted for publication in Nature Communications in its current form.

Reply to Reviewer Comments on Nature Communications Manuscript NCOMMS-18-02944-T

Title: “Direct observation of exciton–exciton interactions”

Authors: J. Dostál, F. Fennel, F. Koch, S. Herbst, F. Würthner, and T. Brixner

We thank all reviewers for their detailed comments that helped us improve the clarity of the presentation in the revised version of our work. We have addressed all points in detail as listed below, and we have modified the manuscript accordingly.

Note that Reviewers #1 and #2 provide partially opposing views in their comments: While Reviewer #1 considers the discussion too technical in the simulation section, Reviewer #2 would rather see an even more elaborate theoretical modeling. Since it is impossible to fulfill simultaneously both requests, we conclude that our approach lies somewhere in the middle between both viewpoints and should thus be appropriate in general. We take into account both reviewers’ concerns in the revised version, however, by following the Editor’s suggestion “to provide a short discussion of the limitations in the approach in the main text, as suggested by reviewer #2, rather than removing it entirely.”

Reviewers’ comments in this reply letter are printed in black, our response in blue, and explicit changes to the manuscript in red font. All modifications are also visible in a marked-up version of the manuscript. Whenever page numbers are provided in our reply, they refer to the revised, marked-up version.

We hope that with these extensive responses and modifications, our manuscript can now be recommended for publication.

Reviewer #1

In this manuscript, the authors present a novel interpretation of 2D pump-probe spectroscopy, where the use of high powers gives access to fifth order phenomena. The primary results claimed are (i) the characterization of exciton-exciton interactions and (ii) the possibility of reconstructing the third order response 'cleaned up' by fifth-order contamination.

Overall, the paper is well written, the contents are presented in an ordered way, and almost all the results are credible. The paper undoubtedly deserves publication, but I am not sure that in the present form the paper is suitable for Nature Commun.

I have two main concerns.

1) First, the paper is very technical, and I believe that in few parts the reader could be easily lost. Below more details about that:

1.a) the paper is not self-consistent and heavy use of SuppInfo is necessary to follow the contents described in the main text. I understand the space restrictions of Nature Comm, but maybe a paper of this kind should be published somewhere else in a full article form, with part of the SuppInfo integrated into the main text.

In order to take this comment into account we were aiming for an extended discussion of our results in the main manuscript. In this regard, we integrated the complete section describing the EEI2D signal of direct double excitation originally present in the Supplementary Information into the main text on pages 8-10, including (new) Figure 2 in order to increase the self-consistency and readability of the paper:

“Another possible source of an EEI2D signal [...] properties of the sample and the experimental conditions.”

1.b) the 'Theoretical concept' section starting on page 3 is very technical. Indeed, while the Feynman diagrams formalism is nowadays well-known and largely employed in the ultrafast spectroscopy community, I believe that this part results pretty heavy and awkward to follow for a non-specialist reader and for the broad readership of Nature Comm. This is a second reason why I would suggest the publication on a more specialized journal, which would also give the possibility of integrating part of the SuppInfo in the main text.

We feel that any newly introduced theoretical or experimental concept requires precise formulations and analysis. In our case Feynman diagrams are an ideal tool to approach the challenges of 2DEI spectroscopy. As the reviewer states, Feynman diagrams are “well-known and largely employed”, which is why we also use them here. Indeed, they offer an intuitive (because pictorial) representation of the complex mathematical background. The alternative, writing down the corresponding integrals over the response-function and electric-field pulse contributions, is in our opinion much more formal and less understandable to the average reader. This is why we stick with the present approach based on the Feynman diagrams. This approach is precise and equivalent to the mathematical formulas without having to spell them

out explicitly. In order to appeal to the general reader who is not familiar with all the details of Feynman diagrams, however, we have added a brief explanation to how these diagrams should be interpreted on page 5 and cited appropriate references:

“Each diagram as given in Fig. 1 graphically represents a single possible time-ordered sequence of interactions of the system (described by the density matrix) with the electric field at a given order of the perturbation theory. The current state of the system is represented by the matrix element located in the center of the diagram, the electric field interacting with the system’s “ket” or “bra” vector is given by arrows located on the left or right side, respectively. Time flows from the bottom to the top.”

As a main change to the manuscript we have now included two new paragraphs on pages 4-5 that illustrates the main aspect of the method on a qualitative level, before starting with the detailed analysis:

“Before providing a formal treatment of EEI2D, we describe the essence of the technique on a more intuitive level. Coherent 2D spectroscopy is a generalization of transient absorption spectroscopy in which the sample is excited by a coherent pair of pump pulses instead of a single one.^{7,12} The spectrum of such a coherent pulse pair necessarily contains interference fringes, the density of which can be reliably controlled by the time separation between the constituting pulses (so-called coherence time). All processes subsequently observed in transient absorption can thus be related to the excitation of the sample at a set of specific frequencies (the maxima of the fringes). By systematic scanning of the delay between the pulses, we periodically modulate the excitation of the sample at any given frequency and the excitation at each frequency is modulated with a different rate. If the transient change of absorption scales linearly with excitation power, we can uniquely relate each component of the transient absorption signal to a single excitation frequency (“single-excitation-frequency component”) by observing its periodical appearance and disappearance in the overall signal. In practice this is done by taking the Fourier transform with respect to the coherence time. The resulting signal is named third-order “absorptive” 2D spectrum and appears in the spectral region covered by the pump pulses. Another signal component appears around the origin of the excitation frequency axis (corresponding to that part of the signal not sensitive to scanning the coherence time).

Multiphoton processes in the sample (such as, e.g., any interaction between two excited systems or pump–dump process) make the scaling of the transient change of absorption non-linear with excitation power, which has to be described by (predominantly) fifth-order perturbation theory. In this case, if the 2D spectrum is acquired as described above, Fourier transforming over the coherence time cannot separate the single-excitation-frequency components of the transient absorption signal from each other anymore. The changes in the 2D spectrum are two-fold: Firstly, the absorptive 2D spectrum is overlaid with an additional fifth-order signal, which complicates its physical interpretation. Secondly, an additional signal appears in the new region of twice the excitation frequency, where no single-exciton third-order contribution is present. It can be characterized by $-2\mathbf{k}_1 + 2\mathbf{k}_2 + \mathbf{k}_3$ and $2\mathbf{k}_1 - 2\mathbf{k}_2 + \mathbf{k}_3$ phase-matching relations and is of high interest since it carries solely information about two-exciton

processes. This signal provides valuable direct insight into exciton–exciton interactions fully separated from single–exciton dynamics.

After this qualitative explanation, we now analyze more formally the general properties ...”

We also added the following sentence on pages 6-7 to point out to the non-specialist which features in the Feynman diagrams are specific to EEI2D:

“The particular feature of EEI2D spectroscopy is that processes which are described with Feynman diagrams containing the state $|11\rangle$ can be resolved as a distinctive feature at twice the excitation energy, whereas conventional third-order techniques are limited to diagrams with single-chromophore excitations only (or, to be more precise, where the full ensemble dynamics can be described as if only a single chromophore had been excited).”

1.c) Along the same line, the discussion on page 5, where the authors discuss annihilation, exciton-exciton interactions and how this is manifested or not in 2D signals and the possible cancelation of diagrams is very cumbersome.

We reformulated the mentioned part of the text currently present on page 7 which now reads:

“The assumption of independent population dynamics of both subsystems (decay of states $|01\rangle$ and $|10\rangle$) necessarily fully determines the lifetime and decay pathways of the state $|11\rangle$. For example, in the simplest case of single-exponential population dynamics the decay of state $|11\rangle$ follows the combined rate of both constituting subsystems. This physical requirement assures the perfect cancellation of the overall EEI signal at any time after excitation in case of non-interacting subsystems.

In case the excitons interact with each other the population dynamics of $|11\rangle$ is modified and no longer given by the behavior of the individual constituting subsystems. We rather observe a new, independent physical process, and the associated set of Feynman diagrams does not cancel anymore.”

2) About the reliability of the conclusions and claims:

2a) While the authors thoroughly convinced me that EEI2D could really capture qualitatively exciton-exciton annihilation and characterize exciton diffusion and dynamics, I am not sure that these data can be used quantitatively to extract the exciton parameters reported in Table 1. The calculation of these parameters is based on a model (very complicated, I have to say) relying on a numerous set of fitting parameters. I am not convinced that, also considering the quality of the data and the noise level, there is just a unique set of parameters fitting well the data. The author should comment more on this point and provide additional proofs.

We are happy to hear that we convinced Reviewer 1 (and the two others as well) that EEI2D captures the dynamics of exciton-exciton annihilation, since this is the main point of our work. The EEI2D signal is of fifth order in perturbation theory, compared to the third order of conventional 2D spectroscopy. Thus, naturally, the signal-to-noise ratio is somewhat lower in 2DEE. Nevertheless, with the phase-cycling approach introduced here we are able to extract

the exclusive fifth-order contribution, rather than having to disentangle EEI effects from a mixed third- and fifth-order signal as has been done in previous pump-probe measurements. Our light sources and experimental stability are state of the art.

The main concerns of the reviewer are the “numerous set of fitting parameters” and whether the resulting fit is “unique”. We stress here that, contrary to that statement, we employ a set with *only four independent* fit parameters. Considering that even conventional molecular kinetic pump-probe traces are often fitted with four or more independent parameters in the literature, we find it remarkable that we can fit our full data set, i.e., for all time points and for all laser powers resulting in six measurement curves, simultaneously, with the same set of only four parameters.

The small number of fit parameters has already been stated in the initial version of the manuscript and the SI. However, since the fitting formulas appear to be quite complex, it is not immediately obvious that only four parameters are required. Thus we added a sentence that explicitly states that a small number of fitting parameters is enough to describe the data. We thus modified the discussion on page 16 as follows:

“Equations (1) and (2) appear to be quite complex. Note, however, that despite that appearance, they contain only four independent fitting parameters, by which all time traces for the different excitation powers are fitted simultaneously.”

We cannot provide mathematical proof that the fit always yields a unique set of parameters (i.e., some manifestation of complete linear independence of the parameters.) But this is not possible either for most data fitting applications, and they are still considered useful. We can provide additional arguments, however, why we believe the fit is meaningful and unique:

A) Considering that we use only four fitting parameters to simultaneously describe six curves, each of which has 15 data points, and that each parameter is mostly responsible for one distinct feature (α – overall intensity, τ – overall timescale, β – the long-term curvature and δ – amplitude of the ultrafast rise) it is highly unlikely that identical (or very similar) curves described by (significantly) different sets of parameters exist.

B) The aim of the model is to support the credibility of our technique by showing that the measured time evolution of EEI2D spectra well corresponds to what is expected from annihilation in J-aggregates. Retrieved values of all parameters fall well in the ranges expected for PBI J-aggregates (the only exception is the exciton delocalization length, which is about three times overestimated by the model as stated and discussed in the manuscript).

We have added to the text on page S5 of SI a brief discussion on the uniqueness of the fit as follows:

“The uniqueness of the fit is highly probable since each parameter is mostly responsible for one distinct feature of the fitted curve.”

Additional text is also provided on pages 17 of the main manuscript:

“We thus demonstrated that microscopically meaningful values of fitting parameters can explain the observed strength and time evolution of the EEI2D signal, which validates the EEI2D spectroscopy concept.”

Concerning the reliability of the quantitative results, we further added a description of the limitations of the model directly following the statement above on page 17:

“The detailed description of complicated condensed-phase quantum-mechanical processes in molecular aggregates is beyond the scope of the rather simplistic model. Most notably the dipole approximation used to estimate the oscillator strengths of higher energy states is not fully adequate for tightly packed systems.⁴⁶ This, together with neglecting the influence of higher-excited monomer states,⁴⁷ results most likely in the retrieved rather high exciton delocalization length. Furthermore, a detailed description of early-time dynamics would require addition of purely quantum-mechanical processes such as, e.g., dynamical exciton localization^{48,49} or ballistic propagation.⁵⁰ We hope that the present work will stimulate detailed theoretical treatment of such phenomena in future studies.”

2b) I am even less convinced about the correction of the total signal to retrieve the 'pure' third order signal subtracting the fifth order contributions (claim 2).

The error bar in Figure 5b is so big that it is almost impossible to assess the correctness of the model. In other words, I think that probably a completely different model will perfectly match the data as well. The authors comment about 'small deviations,' but these deviations are in the order of the 80% of the signal!

Also, I do not agree with the conclusion on page 13 where the authors write: 'The match between relative signal intensities is excellent and the shape of data and model coincide concluding that the model is well suited to describe the underlying physical mechanisms. The predicted and reconstructed interaction-free dynamics (Figure 5b) closely resemble each other as well.'

We agree that the noise level in Figure 6b (i.e., Figure 5b in the initial version) seems high. This is to be expected necessarily when considering the error propagation formulas for the subtraction of two (noisy) signals. We emphasize strongly, however, the essential point about these deviations of Fig. 6b: They constitute indeed noise in its true meaning, i.e., *random but not systematic* deviations (except for the short-time behavior of the blue curve which is discussed further in the text). In contrast to that, the curves in Fig. 6a (formerly 5a) very clearly differ systematically from each other. Thus, this proves experimentally that our suggested predictions of the origin of the 2DEi signal are correct because we remove the systematic deviations.

We also respectfully but clearly disagree with the Reviewer's position that “probably a completely different model will perfectly match the data as well”. First, we note from the usage of the term “as well” that the Reviewer apparently agrees that our model indeed matches the data. Second, we stress that, in contrast to the Reviewer's comment, we did *not* assume any

particular model for the annihilation-free decay. Thus it is inappropriate to speculate about whether “completely different models” would also fit, since the data-subtraction process works independently of any particular kinetic model. It is based solely on the perturbation-theory analysis shown in the SI. As long as perturbation theory is an acceptable tool of signal analysis (more on that below in the reply to the next point), the demonstrated data-subtraction procedure should work.

The possibility to correct the data to retrieve the pure third-order signal *without assuming any particular model describing the sample* is one of the novel theoretical predictions of our work. Since this is an important idea we demonstrate it experimentally. In future work, with improved signal-to-noise ratios, it may furthermore become possible to discriminate additionally between various models with assumed different forms of annihilation-free decay, but this goes beyond the scope of the present work which aimed to demonstrate the generality of the approach.

In order to address the comment, we gathered the discussion of retrieval of the annihilation-free decay to a single paragraph in which we reformulated the criticized sentences. The text now states on pages 18-19:

“The presence of annihilation in the absorptive 2D spectra becomes most apparent by normalizing the integrated time evolution by the excitation intensity (**Fehler! Verweisquelle konnte nicht gefunden werden.a**). The decay of the signal is in all cases non-exponential and accelerates with the increasing initial exciton population. Using EEI2D spectroscopy, however, we can reconstruct the interaction-free (without annihilation, direct double excitation and further interaction contributions that were not considered in the model) dynamics by subtracting three times the integrated EEI2D signal from the absorptive transient signal (see Supplementary Information for a derivation). Despite the noise level of the reconstructed curves being rather high, their amplitude and shape (**Fehler! Verweisquelle konnte nicht gefunden werden.b**) resemble well the single exponential decay with a lifetime of (730 ± 90) ps as determined from the fitting model. While the curves in **Fehler! Verweisquelle konnte nicht gefunden werden.a** deviate systematically from each other, the remaining deviations in **Fehler! Verweisquelle konnte nicht gefunden werden.b** constitute purely random noise (except for the short-time behavior of the blue curve), validating experimentally the correctness of our analysis. Note that the reconstruction of an annihilation-free third-order signal works independently of any particular assumed kinetic model. The deviations of the high-energy blue curve are most likely caused by experimental imperfections.”

The idea of merely subtracting 3x the fifth order contribution from the overall signal appears to be very naive and not really complying with the complex dynamics characterizing a complex system at high energy. Without considering that at high powers the perturbative approach on which Feynman diagrams formalism is based is not entirely valid anymore.

The notion to call the subtraction process “very naive” implies that we have made unjustified assumptions. This is not the case. On the contrary, as discussed above, this result appears from perturbation-theory analysis independent of “the complex dynamics characterizing a complex

system at high energy". Thus, to the degree that the system can be described by a combination of third- and fifth-order contributions, the stated result is correct. We believe this to be already a significant result.

It remains to be discussed here, then, whether even higher-order contributions (i.e., starting at seventh order) are also relevant and influence the results significantly. As emphasized above, our described correction is exact for ranges of energies in which the fifth-order terms are much stronger than the seventh-order terms. Such experimental conditions can indeed be realized (see discussion below Fig. R1). This limit, i.e., that fifth-order dominates over seventh-order contributions, is analogous to the assumption throughout virtually all the literature on transient absorption (and 2D) spectroscopy that third-order (single-exciton) dynamics dominate over fifth-order contributions. Without the latter assumption (sometimes checked via conducting measurements with varying excitation power, sometimes not even that), the vast majority of the literature on time-resolved spectroscopy would be wrong. Indeed, our present work shows precisely a systematic approach how to address this issue. So, can one generalize this idea to even higher orders?

Indeed, for high excitation powers it is possible to measure also seventh-order contributions, and to separate them from the third and fifth order, which we have seen in preliminary experiments. Such a signal appears at the triple of the excitation frequency and can be acquired using the same phase cycling scheme as discussed in our paper.

Figure R1: Weak seventh-order signal present at 34 nJ dataset of MEH-PBI.

The seventh-order contribution is visible only for quite high intensities. At the data shown in the manuscript now, it becomes very weakly visible for the highest excitation dataset (Figure R1). It is seen directly from that plot that the seventh-order amplitude is much smaller than the fifth-order amplitude, and thus it is justified in the majority of the analysis to concentrate on third- and fifth-order processes only, especially considering that for the smaller excitation powers the ratio between seventh- and fifth-order contributions will decrease even more.

The final part of our answer to the reviewer's question concerns the point whether one can also reach a regime in which the whole perturbation analysis breaks down. For very high intensities, this may become possible, for example considering processes similar to high-harmonic generation. Such a regime cannot be analyzed with the formalism introduced here, but that was also not the purpose of our study because one can (and we have) deliberately choose a moderate intensity regime where the analysis is applicable. The EEI2D technique delivers meaningful information when this correct intensity range is chosen, and such care in choosing the correct regime is required of any spectroscopic investigation.

We added a corresponding statement on page 19:

“Our results indicate that it is indeed possible to reconstruct the interaction-free decay (which is of third order in perturbation theory) despite the non-negligible presence of fifth-order contributions. The corresponding experimental conditions can be realized with higher excitation power compared to the commonly required case of the dominating third-order single-exciton term, which opens a way to investigate the single-exciton effects of highly interconnected systems.”

Reviewer #2

In this work, the authors describe a new experiment called “exciton-exciton-interaction two-dimensional spectroscopy” which is a fifth order spectroscopy technique that can detect exciton-exciton interactions. In a sense this is rather idiosyncratic, because multiple exciton systems are unlikely to exist naturally. For example, in biological antenna systems natural sunlight is of low intensity and incoherent, and hence is unlikely to produce these types of phenomena.

We thank the reviewer for this important comment because it allows us to clarify the general significance of the EEI phenomenon and EEI2D spectroscopy. Indeed, one might intuitively suspect that “sunlight is of low intensity” and that thus only single-exciton processes are relevant for the function of biological systems. However, experimental evidence shows, maybe surprisingly, that this is not the case.

On a sunny day, the majority of solar energy harvested by a photosynthetic organism is dissipated in the form of heat due to the congestion of the light-harvesting apparatus (see, e.g., Demmig-Adams & Adams, *Nature* 403, 371, 2000, or Krüger & van Grondelle, *J. Phys. B* 50, 132001, 2017). In this situation, multiple excitons interact because the presence of one exciton influences the dynamics and energy-transfer pathways of the other one: The energy of one exciton is being slowly processed in the reaction center while the others have to be quickly dissipated out of the apparatus in order to prevent photodamage. For this purpose, photosynthetic organisms developed numerous protection mechanisms that are still subject to scientific investigation. We believe that such processes might be investigated by our technique as well.

Thus, in conclusion, in that context sunlight is *not* “of low intensity” because the close proximity of chromophores in photosynthesis sets the limit for pure single-excitation dynamics at an extremely low threshold. Incidentally, this is also the reason why studies of time-resolved spectroscopy of photosynthesis are quite difficult because it is very challenging to fulfill the low-light-excitation condition one needs for conventional third-order spectroscopy, and thus very likely many studies exist in the literature where this has not been fulfilled, leading to erroneous results. This was one of the motivations for our present work because it provides a systematic approach to dealing with the higher-order effects.

Thus, we do not consider our technique “idiosyncratic”, because it is directly applicable to an important class of naturally occurring phenomena. There is a second, equally important, reason why EEI2D spectroscopy has a much broader scope: It can be simply considered to be a tool for revealing (single-exciton) diffusion processes despite multiple-exciton excitation. It is not necessary for the considered process to be of “multiple-exciton nature” by itself in order to facilitate a study with EEI2D spectroscopy. The technique is set up to reveal (after suitable analysis as shown in the paper) the exciton diffusion constant, among other properties. Such a constant then signifies the conventional, single-exciton, propagation process. It is irrelevant for the interpretation of the result that the experiment may have required “unnaturally high”

excitation conditions. The resulting outcome after the fitting procedure, nevertheless, displays the behavior as if the system had been excited at low light levels. The key concept here is that we use the novel EEI2D technique to learn something new about the response function of the system (because we employ exciton-exciton interaction to study transport), and that response function is valid independent of the excitation conditions.

We have added a brief statement to make this point clear on pages 2-3:

“The interactions between excitons might be directly connected to the functional properties of multichromophore systems. For example, on a sunny day, the majority of solar energy harvested by a photosynthetic organism is dissipated in the form of heat due to the congestion of the light-harvesting apparatus.^{3,4} In this situation, multiple excitons interact because the presence of one exciton influences the dynamics and energy-transfer pathways of the other one: The energy of one exciton is being slowly processed in the reaction center, while the other excitons have to be quickly dissipated out of the apparatus in order to prevent photodamage. The direct detection of EEI thus might unravel the details of the photoprotective mechanisms.

In other cases, it might be necessary to set the excitation intensity higher than the typical operation conditions of the studied systems in order to induce EEI. Even then the EEI-specific signal carries information about microscopic processes in the studied system (e.g., single-excitation transport properties) that would be relevant for low-light conditions. Detecting of EEI can thus provide new understanding of photophysical processes in a variety of systems.”

Nevertheless, this work is still fundamentally interesting and is an important advance on third order spectroscopy, because it enables one to study situations in which two excitations exist in a system simultaneously. This may therefore open up new directions of research.

Consequently I believe it is publishable in Nature Communications. However, as I explain below, I do have some issues with its present form.

The paper is well written with significant supplementary information. The authors explain the process of interest well, within terms of the traditional double-sided time ordered diagrams. The authors apply the methodology to J-aggregates. This seems an ideal application for this studying exciton-exciton interactions, because of the high conjugation and aggregation properties of these molecules.

In figure 3 of the manuscript, the authors identify the EEI2D signal in the spectra, showing that exciton-exciton interactions can indeed be seen. The authors are also able to extract some dynamical information about the exciton-exciton annihilation process, which is indeed a very exciting result! This shows important promise for future applications for this technique.

We are happy that we convinced the reviewer that the interaction between excitons can indeed be seen directly, since this is the principal novel finding presented in our paper.

The weakness of the paper is the modeling. In my opinion, I think it could be removed and the paper would still be an important contribution. The problem is that exciton-exciton interactions

described in this paper are complicated condensed phase quantum mechanical processes. As the authors admit, the theoretical fits do not satisfactorily fit the experimental data. The authors attribute this to “three- and higher-exciton states”. However, I have significant issues with using simplistic phenomenological kinetic models to describe complex quantum processes.

We agree with the reviewer that processes in J-aggregates are in reality more complex than depicted by our model. The purpose of our model is mainly to convince the reader that the measured EEI signal really captures the interaction between the excitons. For this reason, we chose the simplest possible model (note that Reviewer #1 considers it already very complicated) that reflects all fundamental processes apparent in our data. Even with this simple model we are able to demonstrate that meaningful microscopic parameters (diffusion on the order of few nm²/ps, exciton diffusion over few sites, etc., as expected from other studies on related systems) generate a simulated EEI signal of the exact intensity and evolving on very similar timescales as our experimental data.

We do not admit in general that “the theoretical fits do not satisfactorily fit the experimental data” as the reviewer writes. On the contrary, we claim the fits to work quite well because only four independent parameters suffice to fit simultaneously all curves for the different excitation intensities. A systematic mismatch between model and experiment appears at the highest excitation intensity only (i.e., for the blue curve in Figure 6a, formerly 5a in the initial version). In that case the model is not sufficient. Qualitative reasons for the discrepancy are stated in the manuscript. Please refer also to our discussion of higher-order (i.e., seventh-order) contributions in our answer to a comment by Reviewer 1 on pages 7-9 of this reply letter.

We modified the discussion on page 16 as follows to point out the low number of fit parameters:

“Equations (1) and (2) appear to be quite complex. Note, however, that despite that appearance, they contain only four independent fitting parameters, by which all time traces for the different excitation powers are fitted simultaneously.”

Certainly we would be more than happy if our current work stimulates more detailed quantum treatments of exciton-exciton interaction in the future; the current status of the model demonstrates the essential features of the origin of the EEI signal. Note that we have already implemented more detailed, microscopic, simulations of exciton-exciton interactions for the case of power-dependent pump-probe data (exemplified on a related polymer system in Hader et al., PCCP 19, 31989, 2017). That work is based on a theory by May & Kühn, Charge and Energy Transfer Dynamics in Molecular Systems, Wiley-VCH, 2011. It is possible to apply that microscopic model also to the case of EEI2D spectroscopy. Those results, however, far exceed the scope of the current manuscript and will be published elsewhere (Süß et al., Mapping of exciton-exciton annihilation in a molecular dimer via 5th-order femtosecond two-dimensional spectroscopy, in preparation, 2018).

There are numerous approximations made that the authors are either not aware of, or do not wish to acknowledge. First, they make the dipole approximation, which may well be unsuitable

for exciton-exciton interactions, especially in the condensed phase. (See J. Chem. Phys. 107, 5374 (1997) for details.) The “Initial Exciton Distribution” section of the SI could be more rigorously linked to coherent photonic states (see Molecular Quantum Electrodynamics by Craig and Thirunamachandran or The Quantum Theory of Light by Rodney Loudon). The authors should also look at more rigorous quantum dynamical studies performed on J-aggregates that include explicit quantum processes (J. Chem. Phys. 137, 034109 (2012) and New J. Phys. 16 (2014) 113067), which describe dissipative and QED effects, respectively.

As noted above, the main purpose of the model was to illustrate the emergence of the signal and to validate the experimental approach. We are aware of the approximations that the reviewer rightly mentions. Note, however, that this level of theory based on the diffusion equation we use has been applied previously in the literature to successfully characterize the exciton diffusion in aggregates (see References [28, 29, 39, 40] within the paper). The only novel addition here is the incorporating of the direct double excitation, which was necessary to fully describe the data. The identification of this important process, that was largely omitted in previous analyses, clearly illustrates the benefits of our technique. Advanced modelling as suggested by the reviewer will of course bring a more detailed understanding of the processes in the aggregates, especially combined with possible future datasets with further improved signal-to-noise ratio.

We added a discussion in the main paper on page 17 about the limitations of the model as requested by the Reviewer:

“We thus demonstrated that meaningful values of fitting parameters can explain the observed strength and time evolution of the EEI2D signal, which validates the EEI2D spectroscopy concept. The detailed description of complicated condensed-phase quantum-mechanical processes in molecular aggregates is beyond the scope of the rather simplistic model. Most notably the dipole approximation used to estimate the oscillator strengths of higher energy states is not fully adequate for tightly packed systems.⁴⁶ This, together with neglecting the influence of higher-excited monomer states,⁴⁷ results most likely in the retrieved rather high exciton delocalization length. Furthermore, a detailed description of early-time dynamics would require addition of purely quantum-mechanical processes such as, e.g., dynamical exciton localization^{48,49} or ballistic propagation.⁵⁰ We hope that the present work will stimulate detailed theoretical treatment of such phenomena in future studies.”

Further we added a statement on page 15 about incorporating the direct double excitation:

In order to fully describe the evolution of the EEI2D signal we had to include effects of direct population of the two-exciton state. This effect has not been considered in the previous analysis of transient absorption data^{28,30,39,41–45} since it is not apparent without direct separation of the exciton–exciton interaction part of the signal.

In summary, I believe that this is an important experimental advance, and is publishable in Nature Communications.

We thank the reviewer for this positive recommendation.

However, I don't see what the modeling section adds. It certainly doesn't help to identify a mechanism, but seems more like a curve fitting exercise, which doesn't really work. Ideally the authors should use a more rigorous model that accounts for quantum mechanical effects in J-aggregates. I do understand that such a model is outside the scope of this paper, so the modeling section could either 1) be removed, or 2) deficiencies should be fully acknowledged in the main paper (not in the SI). I believe that this is very important, because in this field especially, it is easy to fall into the trap of over-interpreting experimental data based on questionable theoretical models. I strongly believe it is important that future studies address this head-on for the good of the field.

As already outlined above, we included the simplified model to illustrate what types of information one can extract from EEI2D data and to prove that the general idea of the novel spectroscopy works as expected. Future studies will surely address more details of the dynamic behavior. Concerning the modeling section, we follow the Reviewer's suggested second option above, which is also the recommendation of the editor, and address the limitations of the model in the main paper as indicated above.

Reviewer #3

Dostál et al. detail the innovation of a new ultrafast multidimensional optical spectroscopic technique, exciton-exciton-interaction two-dimensional spectroscopy (EEI2D), to isolate inter-exciton interactions and elucidate exciton diffusion and delocalisation length scales of a model J-aggregate dye system. These key parameters are important to gaining a more fundamental understanding of exciton transport which is key to the primary steps of natural photosynthesis and the field of photovoltaic materials, especially those involving organic photoabsorbing layers.

This well written and thorough manuscript details both the theoretical derivation behind EEI2D and its experimental implementation. These 5th order experiments have been handled with due diligence and care, power dependencies have been performed, alongside control experiments on annihilation free samples. Together this provides a strong rationale and proof that the experimental observations of the J-aggregate MEH-PBI are indeed correct.

The paper will have an impact on how the ultrafast spectroscopic community investigate exciton-exciton interactions, and I expect after the paper's publication and dissemination, the EEI2D technique will be adopted by other leading laboratories around the globe.

It is my very firm recommendation that this manuscript should be accepted for publication in Nature Communications in its current form.

We thank the Reviewer for this clear support.

Reviewer #1 (Remarks to the Author):

I appreciated the efforts of the authors in addressing all the points I have raised and clarifying aspects of the proposed spectroscopy and relative interpretation. I think that the manuscript in this form can be published in Nature Comm.

Just a last comment.

It is true that my requests of 'simplification' could appear in some way in contrast with the request of Referee#2, asking for more detailed theoretical modeling. Nevertheless, I think that, overall, both requests originated from the feeling that something was not entirely clear in data analysis and interpretation: I was asking to simplify, Referee #2 to add details.

The authors certainly improved their explanations, and I think they fulfilled our requests.

While some of the details of the modeling behind the data interpretation still remain not fully clear to me, and I still believe that the data are too noisy to extract 'beyond any reasonable doubt' all the claimed information, I admit that the authors gave consistent and acceptable explanations.

I also think that 'acceptable' is enough in this case, considering that a completely new methodology is proposed, and of course, this represents the first step. I am sure that future work will help to clarify aspects not fully clear right now.

Reviewer #2 (Remarks to the Author):

I have carefully read the authors' responses to my comments, as well as alterations made to the manuscript, and believe this paper is now publishable.

I am still somewhat sceptical about exciton-exciton interactions commonly occurring under natural sunlight conditions. I thank the authors for pointing me to these references, which I will look at.

Nevertheless, that question is outside the scope of this paper, which is now an excellent piece of work that I expect will generate a great deal of interest.

Reply to Reviewer Comments on Nature Communications Manuscript NCOMMS-18-02944-A

Title: “Direct observation of exciton–exciton interactions”

Authors: J. Dostál, F. Fennel, F. Koch, S. Herbst, F. Würthner, and T. Brixner

We again thank the reviewers for their positive evaluation of our work and for all recommending publication in the present form. No further changes were requested. All modifications in the manuscript (see marked-up version) are due to editorial stylistic requests.

Reviewer #1

I appreciated the efforts of the authors in addressing all the points I have raised and clarifying aspects of the proposed spectroscopy and relative interpretation. I think that the manuscript in this form can be published in Nature Comm.

Just a last comment.

It is true that my requests of ‘simplification’ could appear in some way in contrast with the request of Referee#2, asking for more detailed theoretical modeling. Nevertheless, I think that, overall, both requests originated from the feeling that something was not entirely clear in data analysis and interpretation: I was asking to simplify, Referee #2 to add details.

The authors certainly improved their explanations, and I think they fulfilled our requests.

While some of the details of the modeling behind the data interpretation still remain not fully clear to me, and I still believe that the data are too noisy to extract ‘beyond any reasonable doubt’ all the claimed information, I admit that the authors gave consistent and acceptable explanations.

I also think that ‘acceptable’ is enough in this case, considering that a completely new methodology is proposed, and of course, this represents the first step. I am sure that future work will help to clarify aspects not fully clear right now.

Reviewer #2

I have carefully read the authors' responses to my comments, as well as alterations made to the manuscript, and believe this paper is now publishable.

I am still somewhat sceptical about exciton–exciton interactions commonly occurring under natural sunlight conditions. I thank the authors for pointing me to these references, which I will look at.

Nevertheless, that question is outside the scope of this paper, which is now an excellent piece of work that I expect will generate a great deal of interest.